# The Impact of High-Pressure Homogenization and Thermal Processing on the Functional Properties of De-Fatted Chickpea Flour Dispersion

**DOI:** 10.3390/foods12071513

**Published:** 2023-04-03

**Authors:** Zhigang Huang, Jiayi Zhang, Guoliang Zhang, Fei Gao, Chonghao Bi

**Affiliations:** 1School of Artificial Intelligence, Beijing Technology and Business University, No.11 Fu Cheng Road Haidian District, Beijing 100048, China; 2Beijing Key Laboratory of Quality Evaluation Technology for Hygiene and Safety of Plastics, Beijing 100048, China; 3School of Food and Health, Beijing Technology and Business University, No.11 Fu Cheng Road Haidian District, Beijing 100048, China

**Keywords:** defatted chickpea flour, high-pressure homogenization, heat treatment, rheological properties, freeze–thaw stability

## Abstract

Defatted chickpea flour (DCF), a rich source of protein and starch, is frequently utilized in the food industry. Two crucial methods of modifying food materials are high-pressure homogenization (HPH) and heat treatment (HT). This study investigates the effect of co-treatment (HPH-HT) on the particle size, rheological behavior, and thermal characteristics of DCF suspensions. The results indicate that both HPH and HT can result in a more uniform distribution of particle size in the suspensions. The effect of HPH on *G*′ was observed to be reductionary, whereas HT increased it. Nevertheless, the HPH-HT treatment further amplified *G*′ (notably in high-concentration DCF), which demonstrates that the solid properties of DCF are improved. The apparent viscosity of the suspensions increased with individual and combined treatments, with the HPH-HT treatment of DCF12% exhibiting the most significant increase (from 0.005 to 9.5 Pa·s). The rheological behavior of DCF8% with HPH-HT treatment was found to be comparable to that of DCF12% treated only with HT. In conclusion, HPH-HT treatment shows a synergistic impact of HPH and HT on the rheological properties of DCF suspensions, however, it has limited effect on the particle size distribution and freeze–thaw stability.

## 1. Introduction

The increased demand for plant-based protein-based foods in recent years has led to increased interest in pulses as a source of plant-based protein [1]. Chickpeas, in particular, are a popular food source due to their high content of nutrients and bioactive components. The protein in chickpeas is considered to be of a higher quality than that found in other legumes [2]. Defatted chickpea powder (DCF) is a light-yellow powder with higher protein content (18–28%) than other grains and meat. It also contains high amylose content (36–41%), which affects its gelatinization performance [3]. The functional properties of pulses, including chickpeas, are not only affected by the protein and starch content, but also by the interaction between the components under different conditions.

Modification methods, such as heat treatment (HT), can significantly change the functional properties of proteins. HT modification has been widely used to alter the properties of water-soluble proteins because it is convenient to operate [4]. HT treatment increases the nutritional value of pulses, reduces their unpleasant taste, and eliminates anti-nutrients that can be found in pulses [5]. During the heating process, the protein and starch structures in pulses are affected, altering the properties of the flour. Studies have also shown that HT treatment can affect the functional properties of different types of proteins. For example, Jimnez-Munoz et al. found that HT treatment of pea and rice protein dispersions at 90 °C for 15 min, which can improve solubility and form larger aggregates [6]. O’Flynn et al. found that the solubility of soybean protein isolate after HT treatment at 90 °C can change and develop a weak gel-like structure [7]. Peyrano et al. found that HT treatment of cowpea protein isolate at a relatively low temperature can form a good gel (*G*′ > *G*″) and improve stability [8]. Ingrassia et al. revealed that mild HT treatment of defatted soy flour can modify its protein techno-functional properties such as solubility, cold-set gelation, and the aggregation behavior [9]. Meanwhile, HT treatment also affects the starch in plant materials. For example, Hou et al. found that HT treatment can increase the hydrolysis rate of mung bean starch by gelatinizing it [10]. Sui et al. found that HT treatment of cornstarch can lower its peak and final viscosities [11]. Cherakkathodi et al. found that wet HT treatment can reduce the amylose content of modified starch, increase its gelling ability, and enhance the hardness of the resulting gel [12].

The food industry has utilized various modification methods to enhance the quality, safety, and functionality of food products while promoting sustainability. Non-thermal techniques, such as high-pressure homogenization (HPH), have become increasingly popular in achieving these goals. HPH technology, a kind of dynamic pressure treatment, uses the application of mechanical stress, involving turbulence, cavitation, and shearing, to affect the structure of protein and starch and change the properties of fluid and its components, including particle size reduction, solubility changes, interaction modifications, viscosity alterations, and other physical and chemical property changes [13]. Research has demonstrated the impact of HPH treatment on starch. Zhu reviewed the effects of dynamic pressure treatment on starch structure and physicochemical properties and found that it has a significant impact on the rheology and thermal properties of starch [14]. However, a higher concentration of starch tended to give higher viscosity of the suspension, which in turn led to more resistance of the starch systems to the effects of the dynamic pressure treatments. With regards to protein, Zhao et al. studied the effect of HPH pressure level on protein aggregation and structural conformation and found that HPH (the model of the homogenizer was SCIENTZ-150, while the pressure applied from 10 MPa to 80 MPa) can affect the secondary structure of most globular proteins [15]. Guo et al. evaluated the impact of HPH on the structure and emulsifying properties of heat-soluble polymerized kidney bean protein and concluded that HPH (the model of the homogenizer was T-25,93 IKA, while the pressure were 30 and 60 MPa) promoted the formation of disulfide bonds between molecules by hydrophobic interaction, leading to the formation of protein soluble aggregates [16].

The combination of HPH and heat treatment (HT) has been found to offer attractive opportunities for developing novel food-grade biopolymers with improved thermal and structural properties [17]. Sun et al. investigated the simultaneous treatment of prolamin HT and HPH and found that the co-treatment enhanced secondary structure changes and stability [4]. In a review of HPH principles and applications, Levy et al. concluded that HPH can alter the interaction between proteins and starches in an aqueous phase, leading to separation or association [18]. Alvarez et al. studied the effect of high hydrostatic pressure treatment on the rheology and thermal properties of chickpea flour and heat-induced paste and found that high hydrostatic pressure treatment resulted in different degrees of gelatinization in chickpea starch granules [19]. Dynamic pressure treatments were more efficient in modifying starch physicochemical properties compared to high hydrostatic pressure [14]. Therefore, we will combine the treatment with HPH and HT in order to find out the interaction of protein and starch and its impact on functional properties of low-concentration dispersions after HPH-HT co-treatment. This research aims to examine the rheological properties of different low-concentration dispersions after HT treatment, HPH treatment, and the combined action of both, and improve the application potential of chickpea in food industry.

## 2. Materials and Methods

### 2.1. Materials and Equipment

Defatted chickpea flour (containing about 27% protein and 45% starch) was purchased from Shanxi Panier Biotechnology Co., Ltd., Xi’an, China. All chemicals are analytical grade.

### 2.2. Sample Preparation

The following steps describe a process for preparing defatted chickpea flour (DCF) dispersions with varying concentrations and denaturing them through heating or high-pressure homogenization. The 24% DCF dispersion was prepared by dissolving the DCF in deionized water, which using the Cantilever stirrer (IKA Instrument Co., Ltd., Staufen, Germany) stir for 30 min. The dispersion was then diluted to produce DCF dispersions with concentrations of 18%, 12%, 8%, 4%, and 2%.

#### 2.2.1. Single Treatment

HT sample: 100 mL of each of the 12%, 8%, and 4% DCF dispersions mentioned in Section 2.2 was heated in a 95 °C water bath for 30 min, then cooling to room temperature. 

HPH sample: The 24% DCF dispersion was homogenized with a APV-2000 high-pressure homogenizer (Hammertek Fluid Technology Co., Ltd., Beijing, China) under 40 MPa, followed by dilutions to prepare a gradient of DCF dispersions. The homogenizer is positive displacement pumps equipped with a homogenizing valve. The actual homogenization takes place when material is pumped through the homogenizing valve. Regardless of the homogenization pressure, the flow rate remains relatively constant. Figure 1 is the working principal diagram of the homogenizer.

#### 2.2.2. Combined Treatment

HPH-HT sample: The 12%, 8%, and 4% DCF dispersions prepared through HPH were then heated and cooled as described in Section 2.2.1. 

All the samples were stored in a refrigerator at 4 °C overnight to allow for complete hydration of the DCF dispersion.

### 2.3. Visual Appearance

The appearance of the dispersion samples was recorded by taking photographs of samples using a mobile phone. The model of the cell phone was iPhone 12 Pro, ISO was set to 100, shutter speed was 1/50 s, and the aperture was f1.6. All the samples were stored in the refrigerator at 4 °C for 24 h, and stirred evenly with a glass rod before taking photographs.

The photographs were opened with Photoshop, and a circle of the same size was selected to get the average value of the L*, a*, and b* parameters.

### 2.4. Particle Size Distribution

We evaluated the droplet size of DCF dispersion using an Omega LS-609 laser (Zhuhai Omec Instruments Co. Ltd., Zhuhai, China) particle size analyzer at 25 °C. The dispersion should be mixed thoroughly with deionized water as the dispersant to ensure even distribution and eliminate the presence of bubbles or undispersed droplets. The relative refractive index of the dispersion should be set to 1.470, while the refractive index of the dispersant should be 1.330. The agitator speed should be set to 2500 RPM during measurement.

### 2.5. Rheological Properties

All rheological tests were characterized using a DHR-2 rheometer (TA Instruments, New Castle, DE, USA). Referring to the methods used by Bi [20], we actioned some pre-experimental improvements. A 40 mm diameter plate was selected and the distance between the geometry and the Peltier plate was 1000 μm. The Peltier plate was connected to a circulating cooling system. 1 mL of the dispersion was placed on the Peltier plate. Silicone oil was added to the sample using a test tube to prevent evaporation. The experimental data were recorded during the test.

#### 2.5.1. Frequency Sweep Tests

Frequency scan experiments were performed at a frequency in the range of 1–100 Hz. The strain was set to 1% by strain scan (in the range of linear viscoelastic region, which determined by strain scan test) to record the storage modulus *G*′ (Pa) and the loss modulus *G*″ (Pa). *G*′ was used to evaluate the energy storage due to elastic changes, and *G*″ was used to evaluate the energy storage due to viscous changes.

The storage modulus (*G*′) and loss modulus (*G*″) of DCF dispersions can be fitted with a power law function:*G′ = K’**·**ω^n′^*(1)
*G″ = K″**·**ω^n″^*(2)
where *K* is the power law constant (Pa·s*^n^*), *n* is the frequency index, and *ω* is the angular frequency (rad/s).

#### 2.5.2. Steady Shear Tests

The steady-state shear test was recorded using a steady-state flow ramp at a shear rate of 10–1000 1/s, where the temperature was set to 25 °C.

The apparent viscosity of DCF dispersions can be fitted with a power law function:*τ = K**·**γ^n^*(3)
where *τ* is yield stress in (Pa), *K* is consistency index (Pa·s*^n^*), *γ* is shear rate (s^−1^), and *n* is flow behavior index.

#### 2.5.3. Temperature Sweep Tests 

Set the maximum strain of 0.5% and the frequency of 1 Hz for temperature scanning. The temperature was increased from 25 °C to 80 °C at a rate of 10 °C/min and then held for 400 s, and then decreased from 80 °C to 25 °C at a rate of 10 °C/min. The storage modulus (*G*′) and loss modulus (*G*″) were obtained by setting the testing mode to oscillation.

### 2.6. Thermal Properties

The thermal properties were evaluated using Differential Scanning Calorimetry (DSC250, TA Instruments, New Castle, DE, USA). Samples (15 µL) were placed into liquid aluminum pans and the sealed. The sample pan was placed on the electrode outside the DSC chamber, and an empty pan was used as the reference pan on the inside. The nitrogen flow rate was kept at 50 mL/min and the sample was cooled from 40 °C to −40 °C at a rate of 10 °C/min, and then heated back to 40 °C at a rate of 10 °C/min [21].

## 3. Results and Discussion

### 3.1. Visual Appearance Results

Table 1 shows the L*, a*, b* parameters of DCF dispersion with different concentrations. Figure 2 illustrates the appearance of DCF dispersions with varying concentrations, and Figure 3 contains three images of DCF dispersions treated through different methods at different concentrations. As the concentration decreased, for samples with 2% concentration, the value of L* can be seen that the sample color was lighter. When the concentration is higher than 8%, a small amount of 10 mL sample has little difference in color. The appearance of the DCF dispersion treated by HPH is similar to the untreated sample. However, the DCF dispersion treated by HT at a concentration of 12% appears to be gel-like. At 8% concentration, the flow of the dispersion is found to be resistive, but at 4%, the dispersion still has a liquid consistency. It was observed that HPH-HT co-treatment at a concentration of 12% resulted in a higher gel formation compared to the dispersions treated by either method alone, indicating that the combined treatment improved the smoothness of the dispersion.

### 3.2. Particle Size Distribution Results

D_(4,3)_ in Table 2 displays the mean diameter particle size of DCF dispersions that have undergone different treatments, as dissolved in deionized water. The figure shows that the DCF dispersion treated with HPH at 40 MPa is smaller compared to the untreated sample. This is attributed to the homogenization treatment at 40 MPa, which helps preserve the original shape of most of the starch granules [22]. The treatment breaks the hydrogen bonds and disulfide bonds of proteins, resulting in their dispersion into small molecules. On the other hand, the mean diameter of the HT-treated DCF dispersion increases. This is because the heating process mixes the protein and starch in the dispersion, leading to cross-linked protein and starch molecules forming a uniform network [5], thus causing an increase in particle size. Additionally, protein denaturation happens at about 85 °C, which also promotes protein aggregation [9].

The polydispersity index (PDI) reflects the molecular weight distribution of the dispersion [23]. Table 2 shows that the PDI value of the HPH treated sample is lower than the untreated dispersion, which indicates that HPH treatment can lead to a more uniform distribution of molecular weight in the system. The lowest value is found in the dispersion of HT-treated, which indicates the most uniform distribution. The value of the sample after HPH-HT co-treatment is slightly higher than the dispersion treated with HT alone. This may be due to the fact that HPH treatment can destroy the spatial structure of the protein, which increases the aggregation behavior during the heating process, resulting in a higher value of PDI.

### 3.3. Frequency Sweep Tests Results

Figure 4 illustrates the relationship between the modulus and angular frequency with different concentrations of DCF dispersion. The viscoelastic properties of dispersion can be analyzed by measuring the small oscillation frequency sweep, where *G*′ shows the elasticity while *G*″ reflects the viscosity of the fluid. The results show that as the concentration of DCF dispersion increases, both *G*′ and *G*″ increase. *G*′ is the energy stored in the sample due to elastic deformation during deformation, which depends on the number of interactions and the binding energy between protein chains [24]. *G*″ is the energy lost in the form of heat during the flow of the sample due to viscous deformation and represents the viscous behavior of the sample. With the raise of angular frequency, all samples exhibit a steady upward trend in both *G*′ and *G*″, which also have a certain frequency-dependent behavior. This can be attributed to the fact that at higher frequencies, the temporary cross-linked junctions formed by the entanglement of protein molecular chains are not able to be broken during short oscillations [25]. Comparing Figure 4a with b, it can be clearly observed that the values of *G*′ are always higher than *G*″, which indicates that the elasticity of the DCF dispersion is better than the viscosity in the linear viscoelastic area. The same finding was reported by Huang et al. in a study about the rheological properties of peanut protein suspensions [26].

Figure 5a–c display the trends of the storage modulus (*G*′) and loss modulus (*G*″) with frequency under different treatments of concentration 4%, concentration 8% and concentration 12%. As seen in Figure 5a, the sample with a 4% concentration experiences a minimal alteration in *G*′ and *G*″ after treated. Notably, HPH-HT treatment does not make a positive contribution to viscoelastic properties, which indicates that the concentration has a significant impact to the treatment of the sample. Figure 5b reveals that after HT treatment, *G*′ and *G*″ increase significantly while the frequency dependence is decreased, which improves the viscoelastic properties of the DCF dispersion and makes the sample appear weakly gelatinous. At this point, it can be seen that co-treatment HPH-HT has a remarkable effect on the viscoelastic properties of the sample. It can be seen from Figure 5c that HPH treatment decreases the storage modulus *G*′, which may be due to the fact that HPH acts on proteins and destroys the structure, reducing the binding energy between protein chains [24]. Meanwhile, after HT treatment, the growth of *G*′ and *G*″ is more significant, forming a weakly gel-like dispersion, which has more solid-like behavior than the sample with concentration 8%. These observations indicate that the HPH treatment at 40 MPa causes protein unfolding in the dispersion and an increase in linear amylose content, while also degrading the structure of DCF system, but it is not enough to form a hard gel network. Alvarez et al. found that high concentrations of DCF dispersions are more susceptible to high hydrostatic pressure in their study of chickpea flour [19]. We can draw a conclusion that an appropriate high concentration of chickpea flour is more susceptible to the pressure both static and dynamic. The frequency dependence of *G*′ suggests that the gel may consist of non-covalent physical crosslinks [27]. After HPH-HT treatment, the increase in *G*′ and *G*″ of the dispersion suggests a higher entanglement density, which could be due to the HPH treatment leading to the partial unfolding of proteins and exposure of hydrophobic amino acids in the sample. Thereby, we enhanced the hydrophobic interactions and formed more disulfide bonds during heating, which induced gelation and enhanced gel strength [28]. During the heating process, the gelatinization of starch also contributes to the increase of *G*′. It can be found that the viscoelastic properties of HPH-HT 8% were similar to the HT 12% sample. This may be due to the fact that HPH treatment causes the protein spatial structure unfolding and leads to different degrees of cross-linking and aggregation, which is dependent on the protein concentration [29].

Table 3 shows the results of fitting the DCF dispersion data to a power-law model function, which can be seen to fit well for all different concentrations of the dispersion after various treatments. For the HPH treatment samples, there is a slight decrease in the values of *n*′ and *n*″*,* which indicates a reduction of frequency dependence of the viscoelastic modulus. The lower the values of *n*′ and *n*″, the less frequency-dependent the viscoelastic properties are. After HT treatment, both HT 12% and HT 8% samples have a significant increase in *K*′ and *K*″ values, which increase the viscosity of the dispersion and make the property closer to an elastic solid. The combination of HPH-HT treatment results in a significant improvement on the elastic properties of high-concentration DCF samples, which almost lost the dependence on frequency. A study by Ribotta et al. found that pea protein isolate improved the storage and loss modulus of tapioca starch, with the protein–starch gel structure depending on the starch source [30]. This can explain the improvement of elasticity, as the HPH treatment may cause a dispersion of DCF proteins and an increase in amylose, which make it easier to form a structured gel network during thermal gelation and improve the elastic properties.

### 3.4. Steady Shear Tests Results

As seen in Figure 6a, with the rise in concentration, the apparent viscosity of the defatted chickpea flour dispersion steadily increases. Viscosity represents the resistance of the fluid to flow and depends on the interaction between the components. When subject to low shear rates, the apparent viscosity of dispersion first decreases, then slightly rises, which indicates that the suspension displays a shear-thinning or pseudoplastic behavior. As revealed by Figure 6b–d, although the viscosity increases slightly, the apparent viscosity of the samples after HPH treatment changes with shear rate in a similar manner to the untreated samples. The reason may be that HPH treatment can cause the destruction of the protein space structure in the DCF, and the hydrolysis of amylopectin to form more amylose, which results in a longer linear structure that cross-links or aggregates at low shear rates, leading to an increase in the apparent viscosity. This behavior is consistent with the findings of Guo et al. in their study of the effect of HPH on kidney bean protein steady-state shear [16]. However, as the shear rate increases, the dispersion still displays shear-thinning behavior. The apparent viscosity of HT treatment samples decreases with rising shear rate, which reveals that the HT treatment dispersion still exhibits pseudoplasticity. The dispersions at concentrations of 8% and 12% in Figure 6c,d, respectively, which display significant increases in apparent viscosity at low shear rates, showing a weak gel-like behavior, as confirmed by frequency sweep experiments. This increase of viscosity can be attributed to the gelatinization of starch and protein denaturation during HT treatment. As the shear rate increases, the weak gel-like structure is destroyed, resulting the decrease in viscosity. Figure 6 also indicates that the apparent viscosity of HPH-HT co-treated sample increases even further. For the higher concentration dispersion, the weak gel formed after HT is smoother, which may be due to the unchanged granule shape of starch before and after HPH [17], and the expansion of gelatinized starch helping to strengthen the thermally-induced protein gels during the subsequent HT treatment [31], leading to a significant increase in the viscosity of the DCF dispersion after HPH-HT co-treatment.

Table 4 shows the rheological properties of different treatment to DCF dispersion at different concentrations in the same shear rate range. The results in the table show that the fitted curves are in good agreement with the experimental data (*R*^2^ >95% for all samples). All samples exhibited pseudoplastic fluids (*n* < 1) [32]. The table illustrates that the dispersion without heat treatment struggles to fit the same curve at both low and high shear rates, indicating a higher sensitivity to shear rate. As the concentration increases, the value of consistency coefficient K also increases, indicating a more viscous system. The K value of the DCF dispersion after HP-HT co-treatment increased even more compared to the HT sample, which also indicates that HPH treatment has positive impacts on protein denaturation and starch gelatinization of chickpea flour during the HT process. 

### 3.5. Temperature Sweep Tests Results

The temperature sweep test was used to characterize the gelation process of chickpea dispersions with temperature. The storage modulus (*G*′) is an indicator of the elasticity of the gel network. The variation of *G*′ with temperature depicts the formation of the gel structure. The storage modulus *G*′ of the DCF dispersions showed that concentration higher than 4% are always much higher than the loss modulus *G*″ during the variation with temperature, and therefore only *G*′ is discussed. Figure 7 displays the trend of storage modulus *G*′ of DCF dispersion at different concentration with time over 1000 s. It can be seen that *G*′ increased slowly with increasing temperature for all samples with little difference in values, which is due to the beginning of protein aggregation and starch gelatinization to form a gel network. The trend of increasing *G*′ in the 400 s interval at 80 °C is more obvious because the gel strength increases further as more denatured proteins are aggregated and more starches are pasted. During the three processes, *G*′ of low concentration dispersions (4% and 2%) increased in a small range, for which the fluctuation remains between 0.01 to 0.1 Pa. As the low concentration can not be enough to form a gel during heating, this can lead to a minimal increase in the sample modulus with temperature changes [33]. With the concentration increase in the interval from 300 s to 700 s, both *G*′ and *G*″ of the DCF dispersion increase significantly. In the cooling stage, dispersions with concentrations of 8% and 12% show a slow decrease in *G*′ with the decrease in temperature, which is due to the thermally-induced gel partially melting upon cooling. On the other hand, the dispersion with a concentration of 18% shows an increase in *G*′ during the cooling process, which is due to the unfolding of the protein at higher temperatures. As a higher concentration, a well-maintained storage modulus during the heating and cooling may also suggest molecular entanglement between starch and/or protein molecules in the paste [34].

Figure 8a,b show the trend of storage modulus *G*′ and loss modulus *G*″ of DCF dispersion at concentration 8% (a) and 12% (b) with time after different treatment. The samples form weak gels (*G*′ > *G*″*)* in the heat preservation stage after HPH treatment. However, there is still a decrease in G′ during the cooling stage, suggesting that the thermally-induced gel at 80 °C is reversible upon cooling [33] and that the protein is not completely denatured. During HT treatment, proteins have been completely denatured, resulting in protein aggregation. At the same time, HT treatment leads the starch granules to break the hydrogen bonds, destruct the crystal structure, and then pursue expansion [34]. The interaction of the protein with starch increases the *G*′ value. The further increase of *G*′ in the cooling stage suggests that cooling can reinforce the gel structure, making the fully denatured thermal gel irreversible. After HPH-HT co-treatment, the *G*′ increases exponentially, which may be because the protein in the dispersion competes with the hydration of the starch, limiting the swelling of starch granules and preventing water molecules from entering the starch granules [35]. After HPH treatment, the destruction of protein molecule structure reduces the confinement ability of starch, leading to the strength of the weak gel being higher in the thermal gelation process.

### 3.6. Differential Scanning Calorimetry Results

Figure 9 illustrates the cooling–heating cycle heat flow curves of DCF dispersions in different treatment with a concentration of 4%. The figure reveals that the four curves exhibit prominent exothermic and endothermic peaks during the cooling–heating cycle, which indicates the crystallization and melting of the dispersion, respectively [36]. It can be observed that the release heat during cooling can raise the ambient temperature, which is due to the starch hydrolysis into dextrin. The DCF dispersion after HPH treatment has a slightly larger exothermic peak and a slightly higher temperature increase during cooling, which may be due to the smaller molecules form from the HPH treatment of the starch, make it easier to hydrolyze and generate more dextrin. The HT treatment sample increases the area of the exothermic peak during cooling, facilitates crystallization and leads to a significant temperature rise. At the same time, the area of the endothermic peak decreases during heating, which can be attributed to the HT treatment. During the thermal process, more disulfide bonds are formed in the dispersion, as well as more denaturation of the protein, which reduces the energy required for heating, thus reducing the endothermic peak area [28]. Unfortunately, HPH-HT co-treatment does not enhance the freeze–thaw stability of DCF dispersions.

## 4. Conclusions

In this study, the impact of HPH, HT, and HPH-HT co-treatment on DCF dispersions were investigated. The results indicate that as the concentration increases, both the storage modulus *G*′ and loss modulus *G*″ of DCF dispersions increase, which lead to an improvement in viscoelastic properties and a rise in apparent viscosity. HPH treatment can result in a more uniform particle size distribution and a slight increase in apparent viscosity, with the effect becoming more pronounced at higher concentration levels. HT treatment causes the formation of a weak gel structure at concentrations above 8%, which results in a significant increase in *G*′ and apparent viscosity. Additionally, the stability of samples improves with the gel structure, becoming stronger upon cooling. The HPH-HT co-treatment improves the rheological properties of the DCF dispersions but has a negative impact on the freeze–thaw stability. 

In a word, HPH treatment improves the uniformity of particle size and the viscoelastic properties of DCF dispersions, while HT treatment results in a weak gel structure which improves the stability and apparent viscosity. The HPH-HT co-treatment improves the rheological properties of the dispersions but negatively impacts the freeze–thaw stability.

## Figures and Tables

**Figure 1 foods-12-01513-f001:**
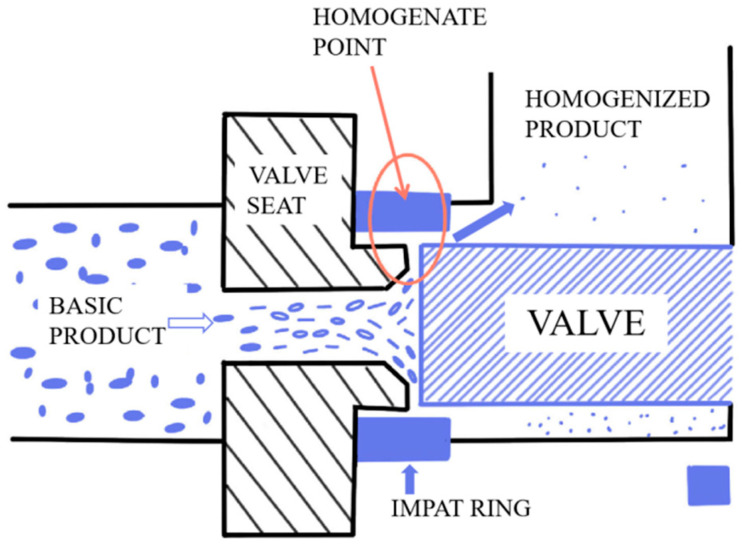
Working principal diagram of the homogenizer.

**Figure 2 foods-12-01513-f002:**
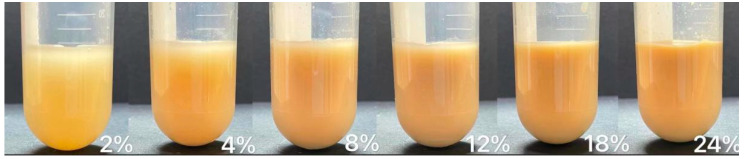
Appearance picture of DCF dispersion with different concentrations.

**Figure 3 foods-12-01513-f003:**
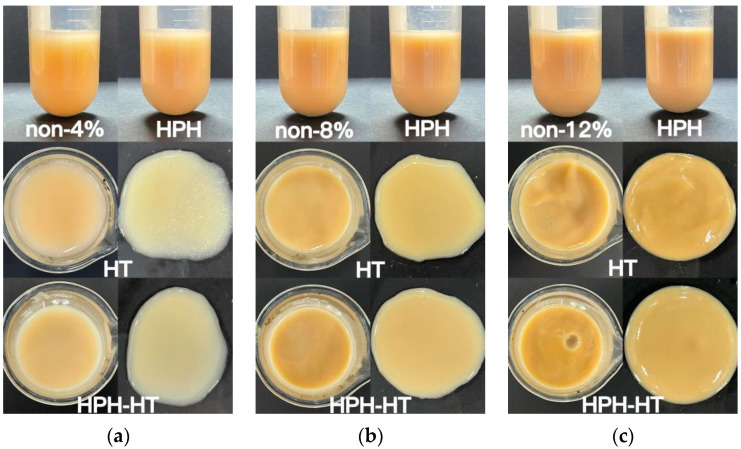
Appearance picture of DCF dispersion treated in different ways with the concentration of 4% (**a**), 8% (**b**), and 12% (**c**).

**Figure 4 foods-12-01513-f004:**
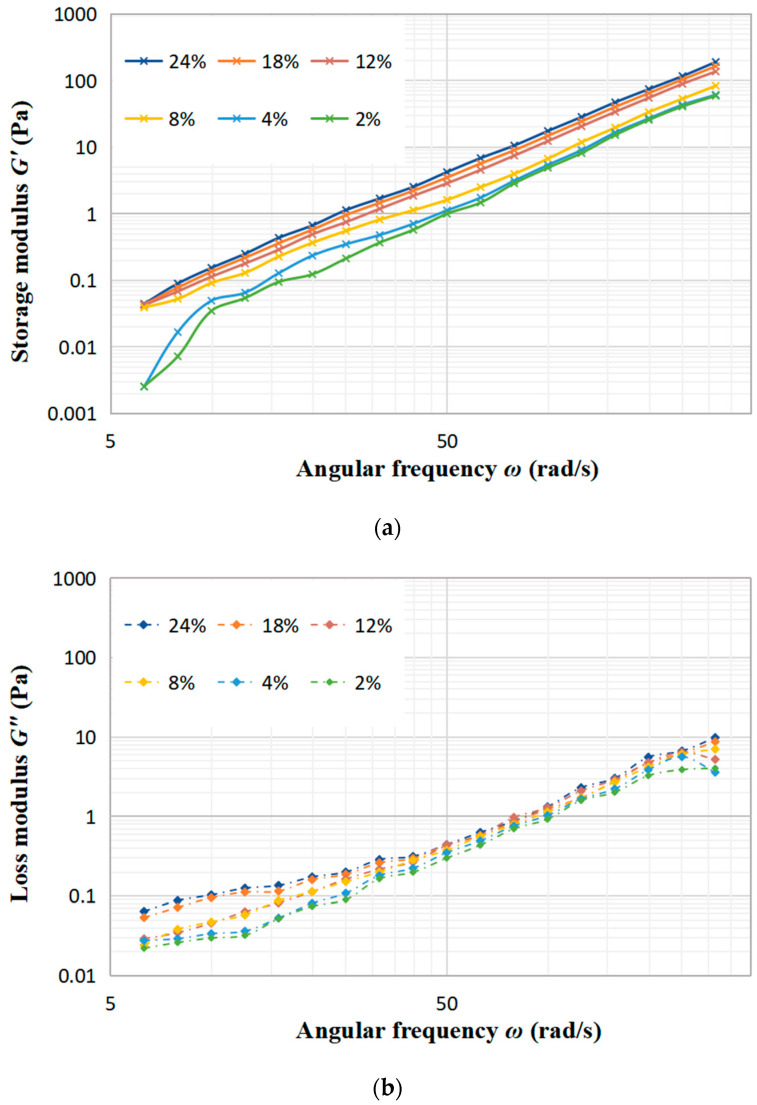
Effect of the angular frequency on the storage modulus *G*′ (**a**) and the loss modulus *G*″ (**b**) at different concentration of DCF dispersion.

**Figure 5 foods-12-01513-f005:**
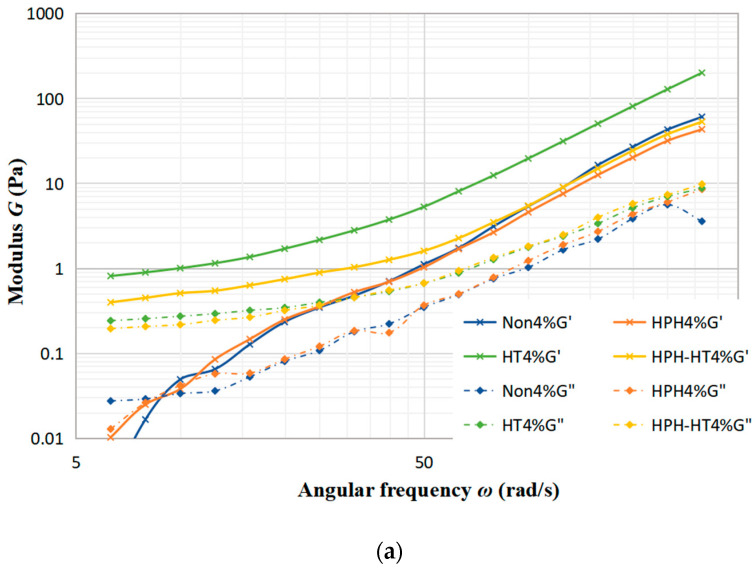
Trends of storage modulus *G*′ and loss modulus *G*″ with frequency under different treatments at various concentrations of DCF dispersion.

**Figure 6 foods-12-01513-f006:**
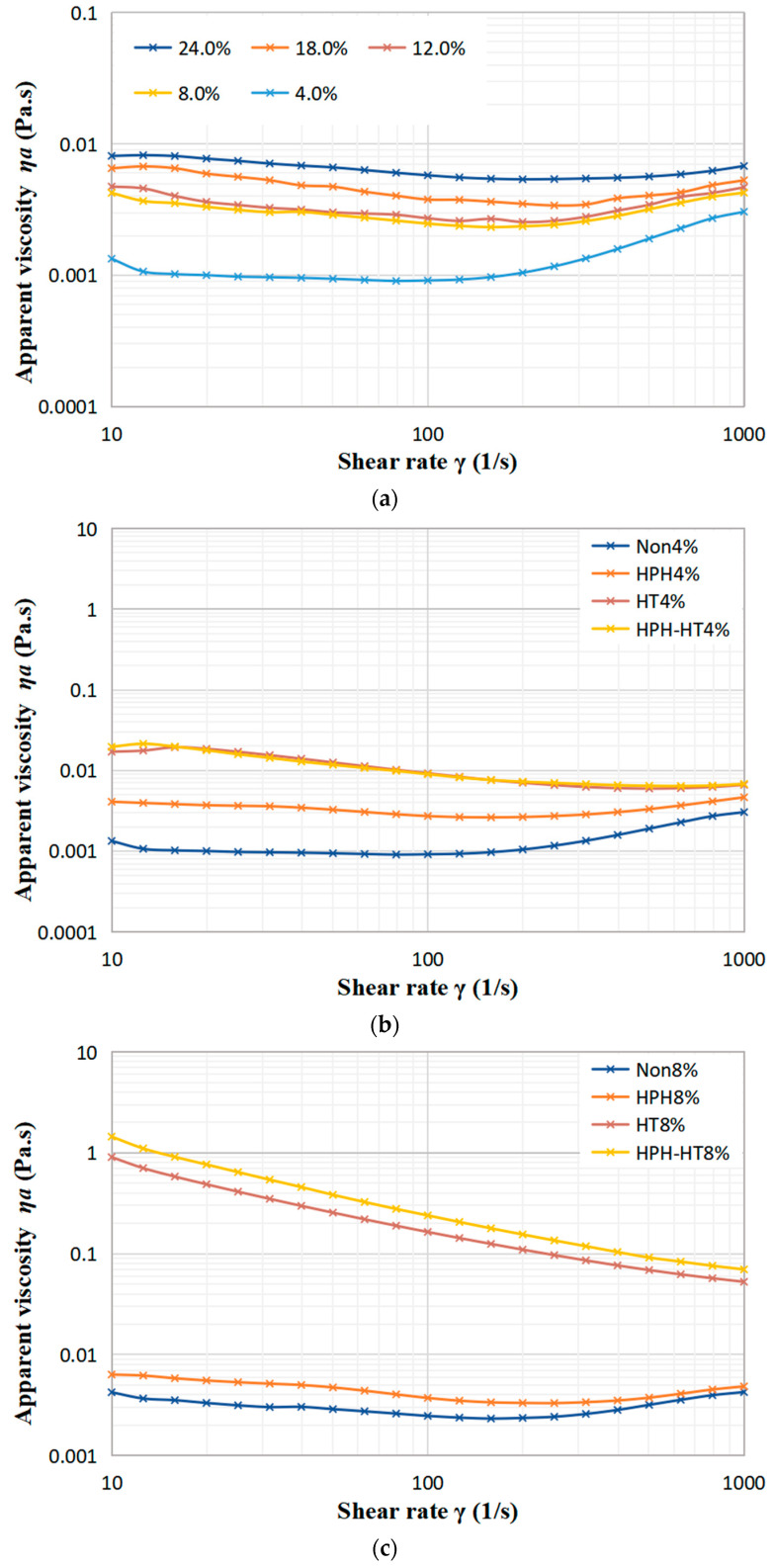
Variation of viscosity of DCF dispersions with shear rate for different concentration (**a**), and different treatment of concentration 4% (**b**), 8% (**c**), and 12% (**d**).

**Figure 7 foods-12-01513-f007:**
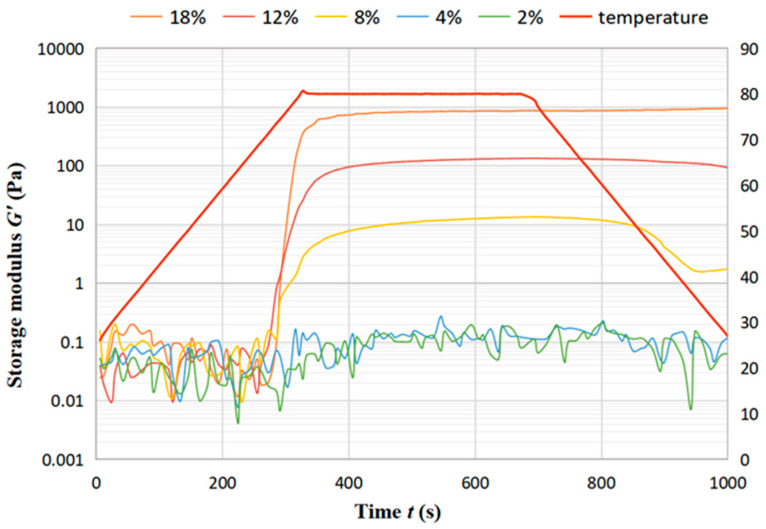
Trend of storage modulus G′ of DCF dispersion at different concentration with time.

**Figure 8 foods-12-01513-f008:**
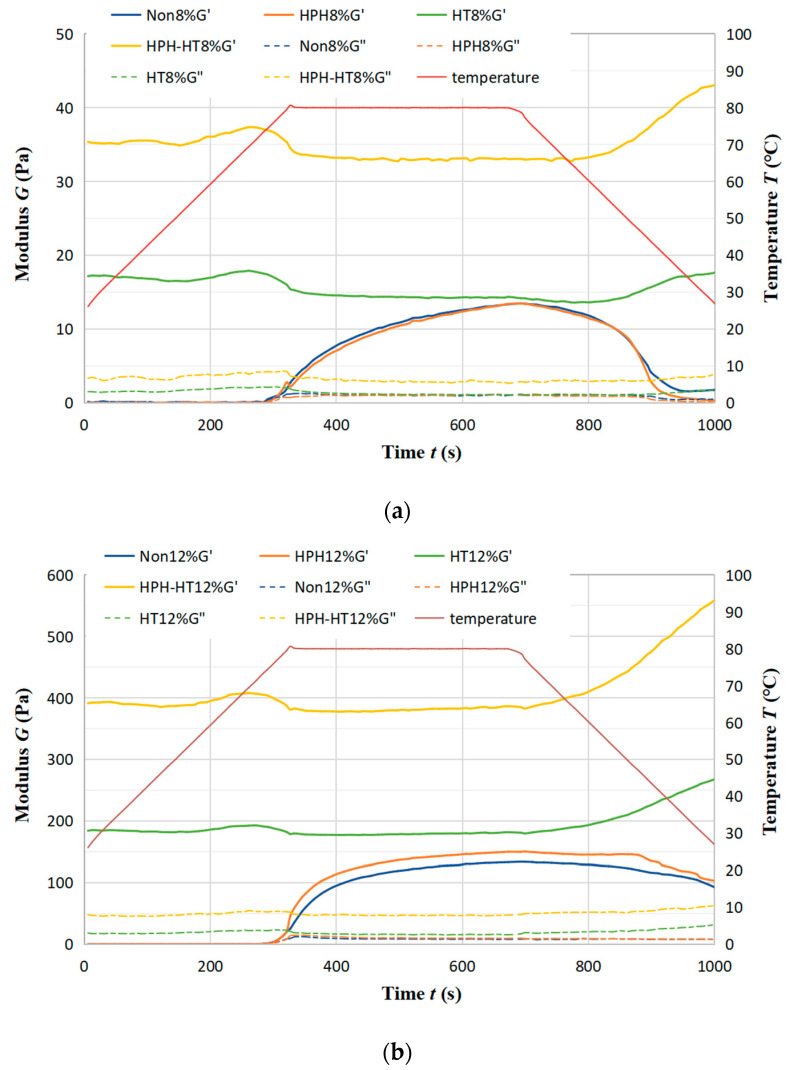
Trend of storage modulus *G*′ and loss modulus *G*″ of DCF dispersion at concentration 8% (**a**) and 12% (**b**) with time.

**Figure 9 foods-12-01513-f009:**
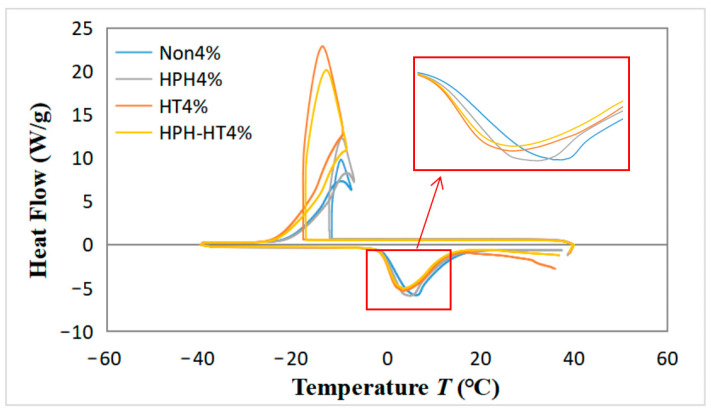
Differential scanning calorimetry (DSC) curves of DCF dispersions in different treatment with a concentration of 4% in one cooling and heating cycle.

**Table 1 foods-12-01513-t001:** L*, a*, and b* parameters of DCF dispersion with different concentrations.

Concentration	2%	4%	8%	12%	18%	24%
L*	80.00 ± 2.94 ^a^	73.33 ± 2.49 ^b^	66.00 ± 1.63 ^c^	65.33 ± 1.70 ^c^	65.33 ± 0.94 ^c^	63.67 ± 3.09 ^c^
a*	10.00 ± 0.82 ^c^	19.33 ± 0.47 ^a^	18.00 ± 2.16 ^ab^	15.00 ± 1.41 ^b^	15.33 ± 1.25 ^b^	17.00 ± 1.63 ^ab^
b*	55.00 ± 1.63 ^a^	49.67 ± 2.62 ^a^	37.67 ± 1.70 ^bc^	33.67 ± 2.05 ^c^	39.33 ± 3.40 ^bc^	40.00 ± 2.16 ^b^

^a,b,c^: Values in a column with different superscripts were significantly different (*p* < 0.05).

**Table 2 foods-12-01513-t002:** Mean diameter D_(4,3)_ and polydispersity Index (PDI) of DCF dispersions treated in different ways.

Sample	Non4%	HPH	HT	HPH-HT
$D_{\left(4,3\right)}$(μm)	44.897 ± 0.043 ^c^	28.410 ± 0.063 ^d^	61.489 ± 0.038 ^a^	51.634 ± 0.116 ^b^
PDI	0.1256 ± 0.0025 ^a^	0.1140 ± 0.0009 ^b^	0.0185 ± 0.0002 ^d^	0.0259 ± 0.0002 ^c^

^a,b,c,d^: Values in a column with different superscripts were significantly different (*p* < 0.05).

**Table 3 foods-12-01513-t003:** Effects of different concentrations of defatted chickpea flour dispersions on power-law model function fitting under different treatments.

Sample	*G*′ *= K*′*·ω^n^*^′^			*G*″ *= K*″*·ω^n^*^″^		
	*K*′	*n*′	*R* ^2^	*K*″	*n*″	*R* ^2^
	Pa·s*^n^*		%	Pa·s*^n^*		%
Non4%	0.001 ± 0.000 ^d^	1.941 ± 0.054 ^ab^	0.996	0.000 ± 0.000 ^d^	1.812 ± 0.038 ^d^	0.998
Non8%	0.001 ± 0.000 ^d^	2.086 ± 0.022 ^a^	0.999	0.002 ± 0.001 ^d^	1.475 ± 0.071 ^abc^	0.987
Non12%	0.001 ± 0.000 ^d^	2.031 ± 0.019 ^ab^	1.000	0.001 ± 0.000 ^d^	1.665 ± 0.037 ^a^	0.997
HPH4%	0.001 ± 0.000 ^d^	1.830 ± 0.052 ^ab^	0.996	0.001 ± 0.000 ^d^	1.647 ± 0.028 ^a^	0.999
HPH8%	0.001 ± 0.000 ^d^	1.939 ± 0.026 ^ab^	0.999	0.002 ± 0.001 ^d^	1.397 ± 0.060 ^bcd^	0.990
HPH12%	0.001 ± 0.000 ^d^	1.957 ± 0.018 ^ab^	1.000	0.002 ± 0.000 ^d^	1.424 ± 0.040 ^ab^	0.996
HT4%	0.002 ± 0.000 ^d^	1.995 ± 0.011 ^b^	1.000	0.004 ± 0.001 ^d^	1.357 ± 0.041 ^cd^	0.994
HT8%	33.256 ± 0.509 ^c^	0.110 ± 0.004 ^c^	0.984	0.745 ± 0.174 ^cd^	0.690 ± 0.045 ^e^	0.957
HT12%	272.295 ± 4.587 ^b^	0.080 ± 0.004 ^c^	0.962	18.680 ± 0.508 ^b^	0.300 ± 0.006 ^f^	0.995
HPH-HT4%	0.001 ± 0.000 ^d^	1.845 ± 0.044 ^ab^	0.997	0.003 ± 0.001 ^d^	1.385 ± 0.037 ^cd^	0.996
HPH-HT8%	49.442 ± 2.882 ^c^	0.190 ± 0.015 ^c^	0.926	3.653 ± 0.207 ^c^	0.411 ± 0.013 ^e^	0.989
HPH-HT12%	410.138 ± 1.341 ^a^	0.110 ± 0.001 ^c^	0.999	37.993 ± 0.815 ^a^	0.255 ± 0.005 ^f^	0.995

^a,b,c,d,e,f^: Values in a column with different superscripts were significantly different (*p* < 0.05).

**Table 4 foods-12-01513-t004:** Effect of different treatment on the fitting coefficient consistency factor k and flow behavior index *n* of DCF dispersion at different concentrations in steady state shear.

Sample	Low Shear Speed			High Shear Speed		
	*K*	*n*	*R* ^2^	*K*	*n*	*R* ^2^
	Pa·s*^n^*		%	Pa·s*^n^*		%
Non4%	0.001 ± 0.000 ^e^	−0.062 ± 0.005 ^a^	0.960	2.042 × 10^−5^ ± 0.000 ^e^	0.731 ± 0.021 ^a^	0.996
Non8%	0.006 ± 0.000 ^e^	−0.182 ± 0.006 ^ab^	0.989	0.000 ± 0.000 ^e^	0.386 ± 0.036 ^b^	0.967
Non12%	0.007 ± 0.000 ^e^	−0.195 ± 0.014 ^cd^	0.962	0.000 ± 0.000 ^e^	0.412 ± 0.040 ^b^	0.963
HPH4%	0.006 ± 0.000 ^e^	−0.175 ± 0.012 ^bc^	0.960	0.000 ± 0.000 ^e^	0.380 ± 0.029 ^b^	0.967
HPH8%	0.011 ± 0.000 ^e^	−0.235 ± 0.011 ^cd^	0.981	0.001 ± 0.000 ^e^	0.302 ± 0.024 ^b^	0.968
HPH12%	0.017 ± 0.010 ^e^	−0.270 ± 0.010 ^cd^	0.987	0.001 ± 0.000 ^e^	0.295 ± 0.034 ^b^	0.950
HT4%	0.052 ± 0.004 ^e^	−0.360 ± 0.018 ^d^	0.965	0.052 ± 0.004 ^e^	−0.360 ± 0.018 ^c^	0.965
HT8%	4.604 ± 0.278 ^d^	−0.733 ± 0.021 ^e^	0.992	4.604 ± 0.278 ^d^	−0.733 ± 0.021 ^d^	0.992
HT12%	31.627 ± 1.871 ^b^	0.844 ± 0.021 ^e^	0.994	31.627 ± 1.871 ^b^	0.844 ± 0.021 ^e^	0.994
HPH-HT4%	0.049 ± 0.003 ^e^	−0.348 ± 0.019 ^d^	0.957	0.049 ± 0.003 ^e^	−0.348 ± 0.019 ^c^	0.957
HPH-HT8%	8.134 ± 0.468 ^c^	−0.776 ± 0.020 ^e^	0.994	8.134 ± 0.468 ^c^	−0.776 ± 0.020 ^de^	0.994
HPH-HT12%	53.258 ± 3.026 ^a^	−0.776 ± 0.020 ^e^	0.994	53.258 ± 3.026 ^a^	−0.776 ± 0.020 ^de^	0.994

^a,b,c,d,e^: Values in a column with different superscripts were significantly different (*p* < 0.05).

## Data Availability

Data is contained within the article.

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
