# Peer review of "The Impact of High-Pressure Homogenization and Thermal Processing on the Functional Properties of De-Fatted Chickpea Flour Dispersion"

_foods, 2023, doi:10.3390/foods12071513_

Round 1

Reviewer 1 Report

The main aim of the work and its novelty is not clear. The research steps, methods, findings and discussion should be improved. In addition, more analysis should be performed to clarify the reasons behind increasing storage modulus and changing particle sizes, such as microstructural analysis. There is a huge problem in the English language and structure of the whole text, which should be carefully edited. Besides:

  1. What is the application of the modified DCF with HPH-HT in industry? Does it have any economic advantage or special application, which HPH method or HT method separately doesn’t have? Please add the answer of these questions to the summary of the work to highlight the main aim of the research.
  2. Ln 36 is repeating the information presented in Ln 34. Please summarize the sentences.
  3. The way of citing references in the body of the text is not correct. Please write the references at the end of each sentences in a right format.
  4. There is no consistency in the introduction of the manuscript.
  5. It is more advantageous to clarify HT and HPH methods in the introduction part.
  6. What is the application of the study? It is highly suggested to improve the main aim of the work in the introduction part.
  7.   Language of the text should be improved.
  8. In section 2.1., writers are repeating the word “was purchased”. This kind of word usage and false grammar can be extensively seen in the article. Please edit the text to present an academic manuscript.
  9. References of the methods are missed.
  10. Material and method are not well formulated in terms of research aspects, as well as English language of the text.
  11. Figures in Section 3.1. should be in higher quality (colour changing is not clear). In addition, discussion of the results is missed.
  12. Please present the G’ and G’’ graphs in one diagram, as a result readers are able to compare the firmness of the gels and dispersions at the same time. Or you can show loss factor in another graph to describe how strong the gels were and how they change during sweep temperature test.
  13. Quality of the graphs should be improved."

Reviewer 2 Report

Manuscript Number: foods-2284826

Manuscript Title: The Impact of High Pressure Homogenization and Thermal Processing on the Functional Properties of De-fatted Chickpea Flour Dispersion

 This paper addresses an important and interesting problem about the modification of chickpea flour. The authors investigated the effect of high-pressure homogenization and thermal processing on the functional properties of chickpea flour dispersion. In general, I think this article need major revisions.

 Comments:

1. The authors should add information on the composition of de-fatted chickpea flour, which is necessary to understand the interaction of chickpea composition under high-pressure homogenization and thermal treatment.

2. The de-fatted chickpea flour processing parameters should be complemented in the Materials and Methods section.

3. Because chickpea flour is a mixture of some ingredients, including starch and protein etc, the interaction of protein, starch and other ingredients and its impact on functional properties of low-concentration dispersions after combined HPH and HT treatment is still not clear. This article lacks data to explain the above problem.

Reviewer 3 Report

This manuscript investigated the impact of high pressure homogenization and thermal processing on the functional properties of de-fatted chickpea flour dispersion. The topic is interesting; however, the manuscript has several problems. 

1. Please check the style of citations.

2. L 34; What do the authors mean by "higher quality"?

3. L 36; Specify the range of starch and protein content.

4. L 37; "Higher content of amylose"? higher than what? what are their ranges? Please express more scientifically. 

5. L 106; Please mention other ingredients of defatted chickpea flour, such as carbohydrate (or starch), lipid, ash, etc.

6. Section 2.2; How about the combined method? This section can be divided into tow parts, 2.2.1 (Single treatment) and 2.2.2 (Combined treatment).

7. Mention more details about part 2.3.1. For example, the model of camera, ISO, shutter speed, aperture, etc. Also, it is necessary to compared the visual appearances with L*, a*, and b* parameters. 

8. Have the tests been done with one repetition? Add the "Statical analysis" section at the end of Materials and Methods and compare the means in tables with the significant letters.

9. As mentioned, section 3.1 should be rewritten based on the L*, a*, and b* parameters.

10. L 194; How did the author evaluate the uniformity of DCF dispersion? Only with a test that is measured visually? In my opinion, this is not enough and needs more evidence.

11. Please provide the figures in higher resolution.

Author Response

请参阅附件。

Reviewer 4 Report

The authors present a valuable idea but unfortunately, the article needs major revisions. Below relevant points that should be reviewed and improved are presented:

INTRODUCTION

All cites must be reviewed as they are not correctly cited inside the text. Example: Line 48, (Jimnez-Munoz et al, 2021), is mentioned in brackets and should not be. Also, this specific cite is not correct, please check the spelling of all the cited researchers.

-     Heat treatment (HT) is described along the article as a standard treatment but it is not. These types of treatments usually use different temperatures and times. Due to the relevance of this parameter into the comparison both  values should be described. Also, it is worthy to describe, for instance, the denaturation temperatures of the mentioned proteins.

-   Line 53:  “good gel” is a very subjective description. Please describe it properly, do you mean a structure with a real gel-like structure (G’> G”), etc.?

-        Line 55: which type of pressure: static or dynamic? Which type of equipment is used to apply this pressure?

-          Line 56: instead of “technical functional” use “techno-functional”

-          Line 57: aggregation is a behaviour, but I would not say it is a functionality. Also, what do you mean by “condensation”?

-          Line 72: is the first time that “dynamic pressure treatment” concept is mentioned. It should be described that HPH is also known as dynamic high pressure and the reason. Also is it necessary to describe since the beginning what you mean by HPH, from which pressures?

-          In general in the description more data should be given: applied pressures, homogenizers used (results may vary among the different equipment), temperatures, treatment times, etc.

-          What about the impact of HPH on starch hydrolysis with negative impact on the gelling properties? This should also be mentioned. Improve the review on the literature to check other effects.

-          Line 96: Alvarez et al. did not use HPH but hydrostatic pressure which is a completely different technology. Please review all the references (cite 5, 24, etc)  as both HPH (high pressure homogenization – DYNAMIC high pressure) and HPP (hydrostatic high pressure – STATIC high pressure) are mistaken and therefore the results from the different studies are not properly compared.

-          What do you mean by combine HPH and HT treatments? It is not clear if the inlet temperature of the fluids is high or not. Also, it is taken into account that HPH also increases the temperature? Even the residence time is very low, a temperature increase takes place.

-          The introduction should be deeply reviewed to give accurate information that is really related to the topic to be dealt with.  

MATERIAL AND METHODS

-          An improved description of the used homogenizer is required. Homogenizers from different providers have different design and this affects the results on the techno-functional properties of the processed ingredients. It is very important to understand how this homogenizer works, which is the valve design, etc. Please include a graph explaining this and give more details in the text. Also explain the working pressure of this HPH unit.

-          Regarding the heat treatment, explain why the samples are firstly heated and not homogenized

-          40 MPa cannot be considered a very high pressure.

-          It is not clear how the control samples are prepared.

-          Visual appearance: did you prepare the tubes at the same time and take photographs after same resting times? Which resting times were used? Pleas give a more detailed description to understand if the photos are comparable or not.

-          Particle size distribution: how did you set this refractive index? Did you measure it?

-          Rheological measurements: a more detailed information should be given about how the different parameters were chosen (frequency, heating rate, etc). if possible refer to the methods used by other authors. Does it make sense to use a plate-plate geometry?, due to the liquid aspect of some samples I would recommend to use concentric cylinder.

-          DSc: considering the water content of the samples, is it possible to use a conventional aluminium capsule? Why did you stop at 40ºC?

-          Was the inlet and outlet temperature measured when HPH treatment was applied?

RESULTS

-          Figure descriptions: please give more detailed information of each figure.

-          I miss the study of the properties of non HT treated samples (control samples).

-          In general, the X and Y axes are almost illegible, please modify the font size in all the figures and make the figures bigger.

-          Line 201: I cannot really see how the HPH samples give a narrower distribution. I suggest describing the mean particle size om Table 1.

-          Particle size: discussion of the results should mention the denaturation temperature of the chickpea protein to understand how the different treatments can modify the structure of this protein.

-          Rheological results seem to indicate the factor with major influence is the temperature, why should we apply HPH (40 MPa)? Rheological results are very confusing an it may be related to the selection of a plate-plate geometry instead of using concentric cylinders (for such low viscosities)

-          Table 2. Please make it more visual.

-          Discussion of results is poor.

GENERAL:

-          Experimental set-up should be reviewed to obtain high-quality results.

-          There are serious doubts that the authors are comparing results adequately. They confuse technologies such as PPH and HPH, which are completely different.

-          There are doubts that the use of a pressure as low as 40 MPa can be considered HPH (technology that is capable of up to 400 MPa). English grammar and style should be reviewed.

Author Response

请参阅附件。

Round 2

Reviewer 2 Report

Manuscript Number: foods-2284826

Manuscript Title: The Impact of High Pressure Homogenization and Thermal Processing on the Functional Properties of De-fatted Chickpea Flour Dispersion

The author made great changes to the manuscript. So I recommend that this manuscript can be accepted.

 Comments:

1. Reference marks in the text should be proofread and modified.

2. Line 17810 ° C10

Reviewer 3 Report

The manuscript is now acceptable.

Author Response

感谢您的指导。

Reviewer 4 Report

- Cites are still incorrect

- Line 53, "good gel" repeated, check

- Table 1 and Table 2. Mean, standard deviation and significance missing

- Figure 9. There is an error, check it

Author Response

请参阅附件
